# Design of a patient-centered decision support tool when selecting an organ transplant center

Sauman Chu[1], Marilyn J. Bruin[1], Warren T. McKinney[2], Ajay K. Israni[3,4], Cory R. Schaffhausen [2,3]*

1 College of Design, University of Minnesota, Minneapolis, Minnesota, United States of America, 2 Hennepin Healthcare Research Institute, Minneapolis, Minnesota, United States of America, 3 Department of Medicine, Hennepin Healthcare, University of Minnesota, Minneapolis, Minnesota, United States of America, 4 Department of Epidemiology and Community Health, University of Minnesota, Minneapolis, Minnesota, United States of America

* schaf390@umn.edu

**Data Availability Statement:** Data is provided as S1 Data Supporting Information.

**Funding:** Agency for Healthcare Research and Quality (AHRQ) and Patient-Centered Outcomes

## Abstract

Patients in the US in need of a life-saving organ transplant must complete a long process of medical decisions, and a first step is to identify a transplant center to complete an evaluation. This study describes a patient-centered process of testing and refinement of a new website (www.transplantcentersearch.org) that was developed to provide data to patients who are seeking a transplant center. Mixed methods, including online surveys and structured usability testing, were conducted to inform changes in an iterative process. Survey data from 684 participants indicated the effects of different icon styles on user decisions. Qualitative feedback from 38 usability testing participants informed improvements to the website interface. The mixed methods approach was feasible and well suited to the need to address multiple development steps of a patient-facing tool. The combined methods allowed for large survey sample sizes and also allowed interaction with a functioning website and in-depth qualitative discussions. The approach is applicable for a broad range of target user groups who are faced with challenging decisions and a need for information tailored to individual users. The survey and usability testing concluded with a functioning website that is positively received by users and meets the objective to support patient decisions when seeking an organ transplant.

## Introduction

Patients in the US in need of a life-saving organ transplant must complete a long process of medical decisions, evaluations, and potentially long waits for a transplant. A first step is to identify a transplant center to complete an evaluation. This hospital is responsible for confirming if transplantation is appropriate for a patient and will conditionally place the patient on the waiting list for a donor organ. The choice of a center is influenced by many factors, including distance, reputation, insurance coverage, and relationships with doctors. However, patients

Research Institute (PCORI) K12 HS26379 (C.R.S.), and AHRQ R01 HS 24527 (A.K.I.), and National Institutes of Health's National Center for Advancing Translational Sciences, grants TL1R002493 and UL1TR002494 (W.T.M). https://www.ahrq.gov/ https://www.nih.gov/ https://www.pcori.org/ The funders had no role in study design, data collection and analysis, decision to publish, or preparation of the manuscript.

**Competing interests:** The authors have declared that no competing interests exist.

are often unaware of what choices are available and may decide based on informal recommendations rather than data [1]. In additional, national policy permits patients to join the waiting list at more than one transplant center in different geographic regions to increase the pool of potential donors [2]. The choice of a transplant center can have a significant impact on survival for the patient. This is due in part to variation in patient outcomes at different centers [3, 4]. While patients can view data on the outcomes at each US center to inform decisions [5, 6]; other factors may influence whether a patient is able to join the waiting list in the first place.

After the evaluation, different centers may be more or less likely to accept a specific patient. Centers have different criteria for patients that reflect willingness and expertise to treat complex or higher risk conditions, such as older age or overweight candidates [7, 8]. As a result, a patient might visit a center for an evaluation, and after days of testing, the patient might be declined. The characteristics of the patients who were recently transplanted can provide some insight about the characteristics of patients the center may accept in the future. This information was previously only available in scientific reports, rather than an accessible online search tool. Therefore, patients are often unaware that another center might have different criteria, and that they may be more likely to be accepted at another location [1].

Research is necessary to see how best to empower and equip patients to make an informed decision with potentially severe consequences because the factors impacting this decision are a complex combination of personal preferences (e.g. distance, relationships with doctors) and clinical risks (not being accepted to the waitlist). Patients with potential risk factors (e.g. older age, overweight) face significant barriers to receiving a transplant because they may be accepted at fewer centers. Decision support frameworks suggest a benefit to both explaining the potential limitations on access to transplant depending on characteristics but also to provide data on potential transplant center options. The quality of decisions can be improved by clarifying the decision and providing facts [9] and following established guidelines for public data reporting [10]. With access to patient-specific information, transplant patients can identify transplant centers that better match their unique medical profile and can increase their access to a life-saving organ transplant.

A patient-centered design process promotes meaningful stakeholder engagement during the development of new resources to support decision making [11–13]. This study describes the testing and refinement of a new website that was developed to help patients select a transplant center, using personalized data to compare transplant center options with successful outcomes and a history of treating other patients with similar characteristics (e.g. older age). This project used multiple methods, including online surveys and structured usability testing, to inform changes in an iterative patient-centered process. The research was guided by design principles and informed by user testing and feedback. The outcome and method can serve as a reference for patient-centered tools for communicating medical information and supporting medical decisions.

## Methods

The study included two phases with differing methods to address distinct design tasks. First, an online survey was used to evaluate multiple options for displaying data about whether center outcomes were "better" or "worse" than similar centers. The icon designs (see Fig 1) were developed with a goal of high contrast because a single report would potentially include many rows and several columns of the same icon. The series of icon designs included distinct approaches to convey worse and better "quality" of centers, and also similar styles with differing color palettes. Icons styles included some iterations with a percentage of shaded area to represent a gradient of worse to better, and other icons using a pointer style. The target audience of organ transplant patients includes predominantly middle aged and older patients, and

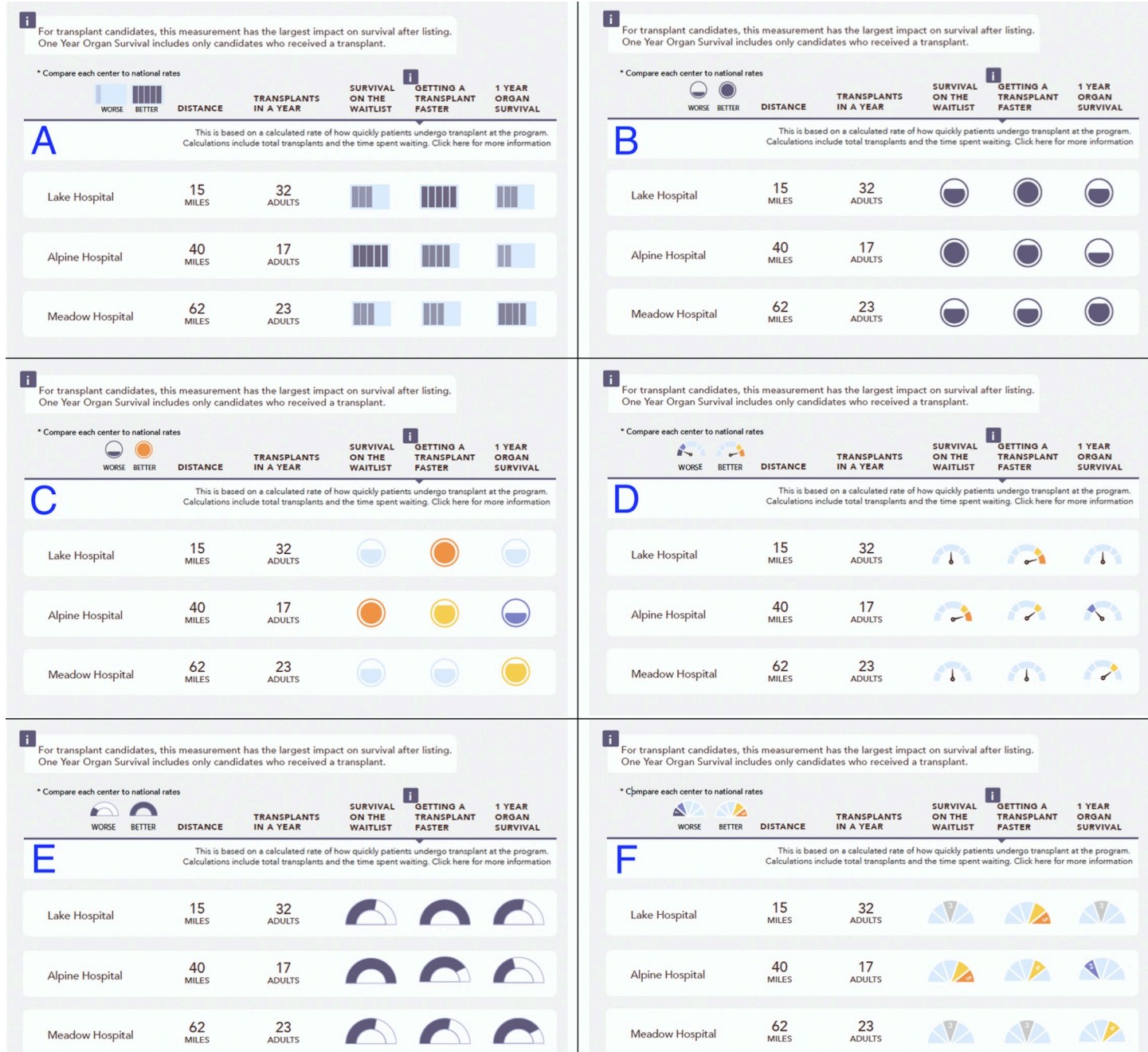

**Fig 1. Images used for the online survey.** Each image included identical hypothetical data but used different icons to represent the same data.

the design used larger font sizes to improve accessibility and readability for all users. The survey allowed for randomized testing and greater sample sizes. Second, several iterations of usability testing with functioning websites were used to refine the navigation, display, and explanations of data for the characteristics of recent transplant patients at a center. The usability testing allowed qualitative feedback and in-depth discussion.

The data provided on the website was obtained from the Scientific Registry of Transplant Recipients (SRTR), a federally funded national data registry for organ transplantation in the US

[14]. The usability testing was conducted using a new website sharing several design elements with SRTR; however, the SRTR site does not include patient specific data and requires federal oversight for changes. A new website permitted flexibility to include additional modifications, including presenting only anonymized center names with sequential numbers (e.g., "Hospital 102") so that real data could be presented for the centers that were providing care to participants.

## Survey methods

The survey study was focused on the icons used to convey the outcome of a center for three metrics: survival on the waitlist, the speed or rate to obtain a transplant, and the probability of survival within 1 year of the organ transplant. Internet users from the United States were recruited online using Amazon Mechanical Turk (AMT) (www.mturk.com, Amazon.com, Inc. Seattle, WA) to evaluate different graphics for display styles. AMT is a platform to post short tasks to an online workforce [15, 16]. Any member of the AMT platform who met the following criteria was eligible to participate: US internet (IP) address, a 95% rate of accepted work on the AMT platform, and at least 50 completed tasks that did not include similar prior transplant surveys. The survey protocol was approved by the Human Subjects Research Committee at Hennepin Healthcare. Each participant consented to the survey electronically; written consent was waived because participants were anonymous. Participants received nominal payments (e.g. $0.50) for the completion of the survey.

The target survey sample size for 6 randomized arms was n = 100 per arm, and the survey was distributed to a minimum of 800 total to account for incomplete or missed screening questions. The sample size was informed by prior studies of randomized surveys suggesting a 0.25 effect size is an appropriate estimate [17].

Surveys were provided in English. A target duration for the survey was 5 minutes. Survey participants answered 5 demographic questions and a series of questions related to choosing a transplant center and interpreting graphics (questions and response choices are shown in S1 Table). Demographics questions included information on whether the respondent had previously been a transplant patient or had family members who were transplant patients. Survey questions were piloted in a previous randomized survey about patient decision making [17]; however, participants from prior surveys were not eligible to complete the survey reported here. Each participant viewed a randomly assigned image depicting a mockup of transplant center search results using differing display styles to depict if a center's outcomes are better or worse than similar centers.

Several questions were exploratory and allowed for open ended responses. Other questions measured impacts on choosing a center and assessed how participants interpreted the graphics. Mockups depicted hypothetical centers and data, including distance and total number of transplants. Question 6 asked participants to select what hospital they would choose given the hypothetical options. Each mockup included a note that Getting a Transplant Faster is the most important factor, and each mockup depicted Lake Hospital with the highest rating for this factor. Therefore, the analysis used Lake Hospital as the best response for this question. Question 9 asked participants to "Review the 1 year organ survival shown for Lake Hospital. Choose the option you believe is most accurate." The 1 year outcome for Lake Hospital represented a center with an outcome in the middle of the range from "worse" to "better." The response options included one choice that was the most accurate interpretation of the graphic.

The survey included an "attention check" question (see question 8 in S1 Table) as a potential method to screen responses that did not carefully read questions. Prior to the final questions, all 6 graphics mockups were displayed together. The survey measured the preferred icon after viewing all choices (see Fig 1).

## Usability testing methods

The preferred graphics and content were incorporated into a functional development website. We then conducted structured usability testing with patients and family members who were seeking a transplant for a kidney, liver, heart, or lung. The usability testing sought to improve navigation and content and to verify if information necessary to identify and compare transplant center options was provided in a clear and usable format. All participants were adults (age 18 or older). The usability tests were conducted at Hennepin Healthcare System (HHS) and the University of Minnesota-Fairview (UMNF) transplant clinics in the Upper Midwest United States. Participants were recruited by research coordinators by phone or by mail. The convenience sample included any English-speaking transplant candidate who was willing to participate. Patients who had agreed to participate had the option to share study details with family members. Patient usability tests and tests with any interested family member were conducted separately. Each participant gave written, informed consent, and the study was approved by the Human Subjects Research Committee at both Hennepin Healthcare and the University of Minnesota (HHS Study #16–4130 / UMNF Study #1606S89161). Participants received a $40 stipend as compensation.

Usability testing with patients was moderated by a trained graphic designer and researcher (S.C). A second researcher (C.R.S.) observed usability tests and moderated sessions with family members. The study was described as an opportunity to collect user feedback about a new patient website with new search tools to review transplant center options. Usability tests were conducted in clinic conference rooms with a laptop used to access the live development website. Participants navigated the website using the laptop and could request assistance if laptop controls (e.g., trackpad) were unclear. Participants followed a discussion guide to complete a series of tasks and follow-up questions (tasks and questions are shown in S2 Table). The participants were instructed to follow a "think out loud" method for cognitive interviews [18]. If needed, participants were reminded to read aloud and verbalize any questions and/or needs. Each usability session was between 30–60 minutes. Usability sessions were audio recorded, and a sample of recorded sessions were transcribed verbatim in order to review comments from early tests as well as later tests that occurred after website revisions. All participants completed a demographics and health history questionnaire (S3 Table).

The usability session progressed through a sequence of pages on the website, in order of the assumed use. Participants first viewed a landing page and introduction. Next, participants entered details about their medical profiles (e.g., age, weight, cause of disease). The website provided additional background to explain why these patient characteristics might be important when considering a transplant center. Lastly, the website displayed a list of options for transplant centers that met the search parameters and displayed data on center outcomes and the characteristics of other transplant recipients at each center.

Usability testing was conducted in separate phases for each organ type, beginning with kidney patients. The website used similar content and navigation for each organ type; however, some organ-specific content was included. As usability testing proceeded, the website content and navigation were revised in response to feedback.

## Results

### Survey results

A total of 811 online participants completed the survey. Of these, 684 participants (84%) gave a correct response to the attention check question. The analysis of responses only included these 684 participants (Table 1). The demographics of the full set and analysis set were similar, with the

**Table 1. Characteristics of participants who completed the survey (Questions #1-#5)** [*].

|  | All Responses | Analysis Set: Excluding Incorrect Attention Check |
|---|---|---|
| Completed surveys; n | 811 | 684 |
| Q1: Age; mean (min / max) | 37 (18 / 80) | 38 (18 / 80) |
| Q2: Sex; % (n) |  |  |
| Male | 47% (378) | 43% (296) |
| Q3: Education; % (n) |  |  |
| Less than High school | <1% (12) | <1% (5) |
| High school | 10% (83) | 11% (75) |
| At least some college | 66% (535) | 67% (460) |
| Graduate education | 22% (181) | 21% (144) |
| Q4: Has had previous transplant; % (n) |  |  |
| Yes | 9% (71) | 2% (17) |
| Q5: Family members needed/received transplant; % (n) |  |  |
| Yes | 40% (322) | 33% (228) |

[*]Some participants left blank responses, therefore totals may not add up to 100%

exception of the proportion who indicated they or a family member had a prior transplant themselves. Table 2 summarizes the survey responses for multiple-choice questions. Questions 6 to 10 include responses after viewing only 1 randomly assigned image. The range of sample sizes per image was a minimum of n = 109 to a maximum of n = 118. Questions 11 and 12 includes a response after the participant viewed all 6 images together (n = 684). Questions 10 and 12 were open ended text responses. These questions resulted in low response rates related to design elements and were similar to responses from usability and previous development phases.

## Usability testing results

A total of 38 participants completed usability testing (see Table 3): kidney (n = 23), liver (n = 7), heart (n = 5), lung (n = 3). Kidney transplantation is the most common and has the

**Table 2. Survey responses for the analysis set: The data consisted of 2 parts.** A first part presented a single randomized image and a series of questions only relating to a single image. A second part presented all images together and followed with a question about all images.

|  | A: Bars | B: Circles | C: Color Circles | D: Color Dial | E: Donut | F: Pie |
|---|---|---|---|---|---|---|
| **Part 1: Randomized** | | | | | | |
| Responses to survey questions after reviewing a single randomly selected image. | | | | | | |
| Q6: Selected best outcome (Lake Hospital), % (n) | 45% (49) | 43% (48) | 48% (55) | 51% (58) | 29% (34) | 53% (62) |
| Q7: Selected most important factor (Getting a Transplant Faster), % (n) | 31% (34) | 26% (29) | 30% (34) | 28% (32) | 22% (26) | 34% (40) |
| Q8: Attention check | N/A. All screening question responses were correct in the analysis set. | | | | | |
| Q9: Accurate Interpretation, % (n) | 16% (17) | 5% (6) | 11% (13) | 43% (49) | 9% (10) | 23% (27) |
| Q10: Describe any information that you do not understand | N/A. Open ended text response. | | | | | |
| Count viewing randomized image | 109 | 112 | 114 | 114 | 117 | 118 |
| **Part 2: Viewing all images** | | | | | | |
| Responses to survey question after viewing all images together, n = 684 | | | | | | |
| Q11: Preferred by user, % (n) | 37% (251) | 11% (72) | 6% (43) | 13% (89) | 16% (108) | 18% (121) |
| Q12: Describe why you selected this icon style | N/A. Open ended text response. | | | | | |

**Table 3. Characteristics of transplant patients and family who participated in usability testing.**

| | Kidney Patients and Family | Liver Patients and Family | Heart Patients | Lung Patients |
|---|---|---|---|---|
| Total Participants (patients and family); n | 23 | 7 | 5 | 3 |
| Patients: Family; n | 20: 3 | 6: 1 | 5: 0 | 3: 0 |
| Age; mean (min, max) | 58 (31, 73) | 64 (56, 71) | 57 (42, 67) | 56 (35, 69) |
| Sex; n (%) | | | | |
| Male | 16 (70%) | 4 (57%) | 5 (100%) | 1 (33%) |
| Race; n (%) | | | | |
| Black | 2 (9%) | 0 (0%) | 1 (20%) | 0 (0%) |
| White | 19 (83%) | 7 (100%) | 4 (80%) | 2 (66%) |
| Asian | 1 (4%) | 0 (0%) | 0 (0%) | 0 (0%) |
| Native-American | 0 (0%) | 0 (0%) | 0 (0%) | 1 (33%) |
| Other | 1 (4%) | 0 (0%) | 0 (0%) | 0 (0%) |
| Education; n (%) | | | | |
| Less than High school | 0 (0%) | 0 (0%) | 0 (0%) | 0 (0%) |
| High school | 6 (26%) | 3 (43%) | 3 (60%) | 0 (0%) |
| At least some college | 15 (65%) | 2 (29%) | 2 (40%) | 3 (100%) |
| Graduate education | 2 (9%) | 2 (29%) | 0 (0%) | 0 (0%) |
| Form(s) of Insurance; n (%) * | | | | |
| Private | 10 (43%) | 5 (71%) | 1 (20%) | 1 (33%) |
| Medicare | 12 (52%) | 5 (71%) | 5 (100%) | 2 (66%) |
| Medicaid | 2 (9%) | 1 (14%) | 0 (0%) | 1 (33%) |
| Not Insured | 0 (0%) | 0 (0%) | 0 (0%) | 0 (0%) |
| Other | 6 (26%) | 0 (0%) | 0 (0%) | 1 (33%) |
| Patient Self-Reported Health Status; n (%) | | | | |
| Excellent | 1 (5%) | 0 (0%) | 0 (0%) | 0 (0%) |
| Very Good | 4 (20%) | 1 (17%) | 2 (40%) | 1 (33%) |
| Good | 10 (50%) | 3 (50%) | 0 (0%) | 0 (0%) |
| Fair | 3 (15%) | 2 (33%) | 2 (40%) | 1 (33%) |
| Poor | 1 (5%) | 0 (0%) | 1 (20%) | 1 (33%) |
| Patients currently on the waiting list, n (%) | | | | |
| Yes | 17 (85%) | 5 (83%) | 5 (100%) | 3 (100%) |
| No | 1 (5%) | 0 (0%) | 0 (0%) | 0 (0%) |
| Not Sure | 2 (10%) | 1 (17%) | 0 (0%) | 0 (0%) |

* Question allowed multiple answers. All questions had less than 5% missing.

greatest number of candidates available to recruit; therefore, most usability testing participants were kidney transplant patients. Usability testing with kidney patients occurred first and included the earliest prototype versions. Any final changes made to the user interface for the kidney search site were also made to subsequent organ types prior to recruiting those participants. For example, the usability testing for the liver search site at the beginning of testing included a user interface that was equivalent to the kidney search site after incorporating revisions from kidney patient usability testing.

Excerpts of usability testing transcripts from kidney patients are shown in Table 4 as a sample of user interface feedback that was addressed during iterations of development and testing. Patients provided critical feedback with both positive and negative responses. Negative responses were considered during iterative revisions to improve usability. Positive comments for important elements of the website design were provided following website improvements.

The usability testing resulted in a change to the scope of information provided on the website. The objective of the study was to provide data to patients who might be declined at some centers due to their characteristics, e.g., older age or overweight. The usability testing included many participants who personally were not outside of typical age, weight, or other criteria. In these cases, viewing data on the characteristics of patients at different centers may not represent a high priority. Future iterations of the website include a sequence of decision guide pages with additional information relevant to a broader demographic group. For example, some centers are more experienced with transplants using living donors or a broader range of deceased donors and these options represent opportunities for patients to learn about different types of donors and to achieve a better outcome through faster access to transplantation.

The interface for the entry page is shown for different stages of development in Figs 2 and 3. Comment boxes appear if the patient enters data outside of typical acceptance criteria, such as an age greater than 70. Fig 4 provides a screenshot of a decision guide page outlining donor types. Fig 5 includes a screenshot of a search results page with data personalized to the user. In the search results, multiple data types are available including outcomes, the characteristics of patients at a center, and donors at a centers. For example, the right-hand column with a pull-down for "recipients criteria" displays a default value based on the user. The user in the image for Fig 5 was a patient over age 70. The search results include outcomes data using icons studied during the online survey.

## Discussion

The studies described here represent a final phase of development prior to providing a live, public website to patients (www.transplantcentersearch.org). The randomized survey informed the selection of graphical icons used to represent patient outcomes data, and usability testing identified areas to improve, such as placement of information and guides to provide context for quality metrics. Several earlier phases informed the development of prototype websites used during the usability testing. Early phases included exploration of multiple design concepts. Based on feedback, some concepts were omitted in later versions, such as patient narratives, and remaining concepts were informed by patient feedback on text and labels. These results are presented elsewhere and do not include usability testing or survey data related to icon design. Previous results include focus groups with transplant patients [1], evaluations of patient decisions when viewing data mockups [17], and mixed methods analyses specific to individual organ types [19, 20]. Previous results also suggest patients may value information that is not available in SRTR data [1]; however, the website options were limited to data that is currently available. As new data become available, search tools can evolve to inform decisions with additional information.

The survey results for feedback on icon styles demonstrated the benefit of the AMT methods and the attention check question. The full set of responses indicated 9% had a prior transplant, a high rate for the general public. The analysis set, limited to correct answers to the attention check question, resulted in 2% indicating a prior transplant. A likely explanation is a higher rate of inaccurate responses among those with an incorrect attention question response, and this discrepancy justifies excluding the data from analysis.

A prior analysis of transplant outcomes indicated that the most important outcome metric for overall patient survival was the measure of "getting a transplant faster" [4], and the hypothetical option "Lake Hospital" had the best outcome for this measure. The survey measured the hypothetical choice of a center to determine how consistent this decision was across the 6 icon display types. The results were similar for different displays, with the exception of option E: Donut. While Style E shared elements with other icons, including monochromatic shading

**Table 4. Excerpts of kidney patient usability testing comments.**

| Comments with rationale for changes and improvements |
|---|
| "One thing I would say is if you can move this [popup message] over here . . . Like and a lot of times like you do a field select . . . you'll get the information pop up like literally on the field itself." |
| Interviewer: Okay, did you read [the popup message]? |
| "I actually did not, I actually kept going down the list to be honest." |
| "Right. So it's a little hard to compare all of them because I lose the header as I scroll down the page, which makes it hard to remember what each column represented." |
| "If you click in the [text entry] box, you have to delete the zero. I'm impatient when it comes to that. But I thought it was pretty easy to find your way around. There weren't a lot of buttons or things to confuse what you were trying to do." |
| "And search for postal code or go over your name. I'm lost on that. What's that question asking? Just my zip code?" |
| "The sentence, compare each center to national rates. The thing is, there's an asterisk there but I don't see anywhere else where there's an asterisk to compare that to." |
| Comments supporting desired usability |
| "I'm going to need a transplant and I'm going to be on dialysis. . . So because of that, this website is very informative for me." |
| "These statistics here [are] the best part about it. . . and I like the little checkmarks." |
| "It was informative. Very informative. It was stuff that I didn't know anything about." |
| "It's very informative and it does give good information but if you're a new patient going on, you get to search for the hospital you prefer. I didn't do this in my first transplant kidney so this is very informative. It's for making a better decision for yourself." |
| "The website would be good for people just finding out they need transplant. This would [have been] good a year ago for me." |
| "I like everything about the website, it is very easy to navigate." |

and left-to-right shading, these elements combined may have been less effective. However, the results did not suggest that the icon style for other options should be selected on the basis of this question.

Question 9 (see Table 2) suggests that the graphic style can impact how users interpret the data on transplant center outcomes. The survey question asked participants to interpret a center with a "middle" rating, e.g. not high or low. A higher proportion of users who viewed the D: Color Dial image selected the correct response for interpreting the outcome shown by the icon. A potential rationale is the increased symmetry of "middle" ratings for style D (as well as style F) which use an icon oriented to the middle as a depiction of a middle rating. However, this image was rated as one of the lowest for preference by the user on Question 11. The contrast in responses highlights a challenge of creating graphics that are engaging and promote the use of a data tool and simultaneously promotes effective interpretation. The websites evaluated during usability testing included the graphic style represented by style A: Bars as this had the highest result for user preference (Q11) and acceptable responses for selecting the best outcome (Q6). The A: Bars style was also adopted by the SRTR site based on previous user preference data, and maintaining consistency was considered an advantage. Additional survey research is warranted to evaluate effective ways to provide context or promote accurate interpretation, such as additional detail for the icon key.

The usability testing provided additional opportunities to refine the website interface for improved usability. Table 4 includes patient quotes relating to poor usability during early testing sessions. The quotes demonstrated that the popup message (see Fig 2) was difficult to read in the original position as a right-side column. The layout was changed to a single column (see Fig 3). Other examples of user interface changes included removing default values from the text entry fields, allowing headers on a scrolling list to remain fixed when scrolling. Minor

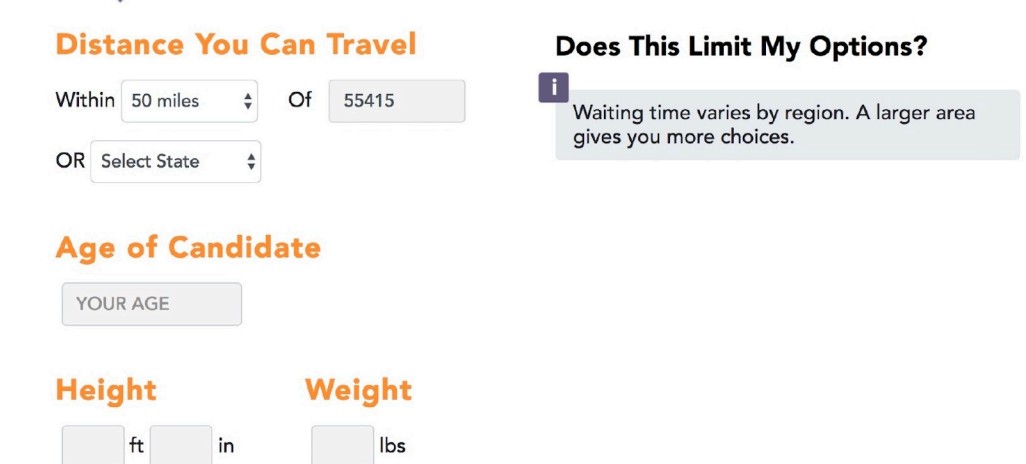

**Fig 2. Development website screenshot of a portion of the data entry page: Initial iteration.** The characteristics shown represent criteria that may impact if a patient is accepted or declined at a center. A pop-up message appears if a user enters data that is outside of typical criteria, such as age greater than 70.

wording issues were addressed, such as switching "postal" code to "zip" code, and to remove unnecessary characters from the icon key.

Later testing sessions included sufficient positive feedback to support release of the tool online (Table 4). The patients were able to navigate pages and could perform the website search in under 20 minutes. In addition, the patients reported learning new information, but were not overwhelmed, potentially increasing awareness of options and how these options may impact outcomes. When answering follow-up questions, patients expressed eagerness to see the website live to support their transplant decisions.

The mixed methods approach was used as a patient-centered design process. Transplantation is a complex treatment option and patients are faced with overwhelming demands to stay informed. The design process and live search tool can serve as a model for other data-driven sites with a specific focus or topic. The search site provides organ specific search tools; however, the site maintains a primary focus on a single decision and a design goal was to solely focus on finding an organ transplant center. The mixed methods approach offered benefits compared to a single method. The design process in many domains can be strengthened by exploration of a diverse range of ideas. User feedback is essential to align a solution with user preferences and needs; however, the small samples common for qualitative feedback do not allow for randomization and statistical analysis. The survey study allowed for a large sample size and the ability to randomize participants across many alternatives. The usability testing was limited to a smaller sample but allowed interaction with a functioning website and in-depth qualitative discussions. The mixed methods approach in the current study were applied to a tool to support patient decisions. The status quo results in patients selecting a transplant center without an understanding of what options may exist, the differences between options, and how these differences may impact their lives [1]. The Ottawa Decision Support Framework is a guide to address decision needs, and a critical need is knowledge related to the decision [9]. The new website provides support that is consistent with this framework and offers information previously only available in scientific reports. The design concepts were also

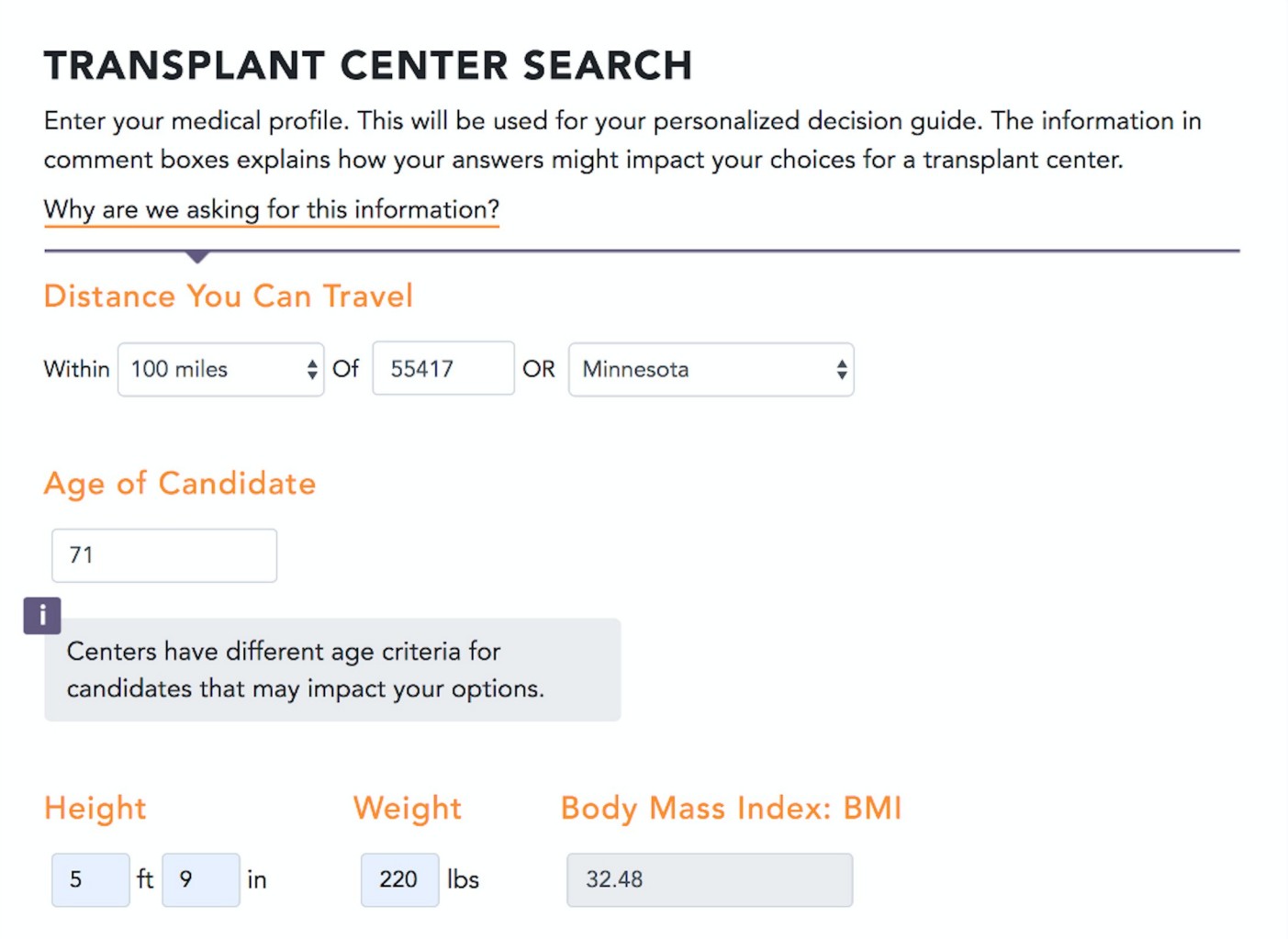

**Fig 3. Development website screenshot of a portion of the data entry page: Final iteration.** The characteristics shown represent criteria that may impact if a patient is accepted or declined at a center. A pop-up message appears if a user enters data that is outside of typical criteria, such as age greater than 70.

consistent with this framework to provide support from others using prominently displayed Print and Email buttons to allow future discussions with family or doctors.

The usability testing and survey data were critical to support patient decisions with the creation of a search tool presenting complex information tailored to individual needs, so patients understand relevant information for their decisions. Qualitative and survey results suggest the website is relevant to patient decisions, and the design was influenced by evidence that the content and style can improve decisions. The website allows user feedback through an online form. Selected feedback from website users is included in S4 Table and indicates positive responses from users and also areas to pursue for future improvements. Future work will also evaluate the effectiveness of the website for patients deciding on a transplant center. A randomized trial is planned to compare the existing SRTR website to the new patient specific website to determine whether the new website with the search tool improves a patient's level of knowledge when seeking a transplant center that has experience transplanting patients with specific characteristics (NCT03610555 on clinicaltrials.gov). Future development will be

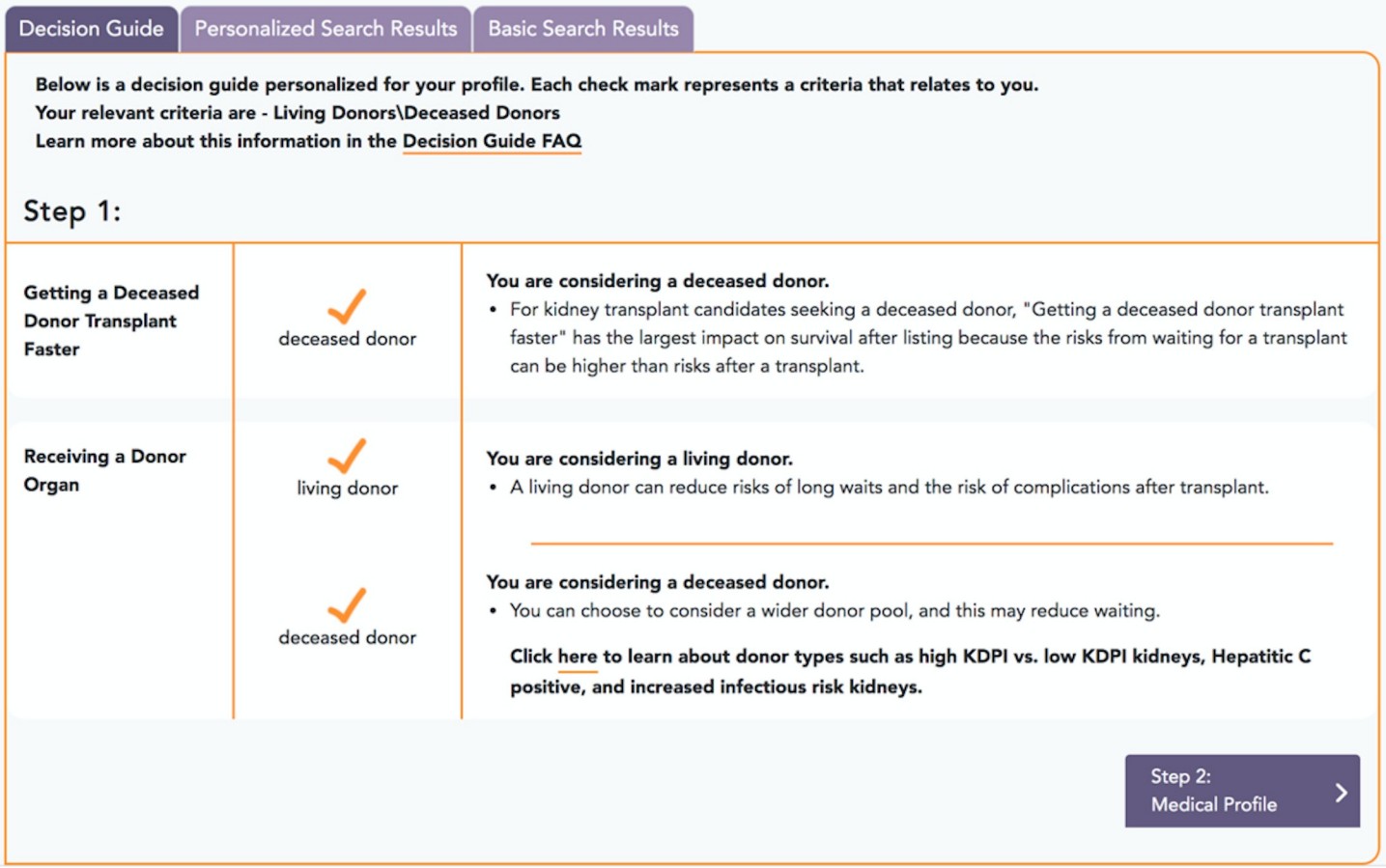

**Fig 4. Development website screenshot of a decision guide to provide context about the transplant process for different donor types.** The guide was added as a resource for users who may not have characteristics that would lead to being declined for a transplant.

influenced by national policy. Recent policy changes for organ allocation have removed geographic boundaries from prioritization of some donor organs [21]. While these changes did not specifically target candidate waitlisting practices that are central to the center search website, patient outcomes at centers may change due to these policies. The website is updated with new SRTR data twice per year and will reflect the most current data available.

While the study included multiple methods to better address the design task, there were important limitations. The survey study recruited participants from the general public and did not limit participants to only those seeking care for a transplant. A survey of only transplant patients would not allow feasible recruiting of large sample sizes. In addition, the target audience for the website included patients who had only recently learned of a need for a transplant and were in early stages of seeking treatment. These patients likely have perceptions of data about transplant centers that are similar to the general public. Family members who may have less direct experience with transplant providers may also assist in decision making. Therefore, data from the general public can inform how information is viewed by patients and family who are early in the process of seeking a transplant center. The usability testing included a sample at a local center. While the sample included multiple organ types, the testing did not include a sample representative of national demographics. The use of non-patients for survey data and a local sample for usability testing may limit generalizability. The website includes a

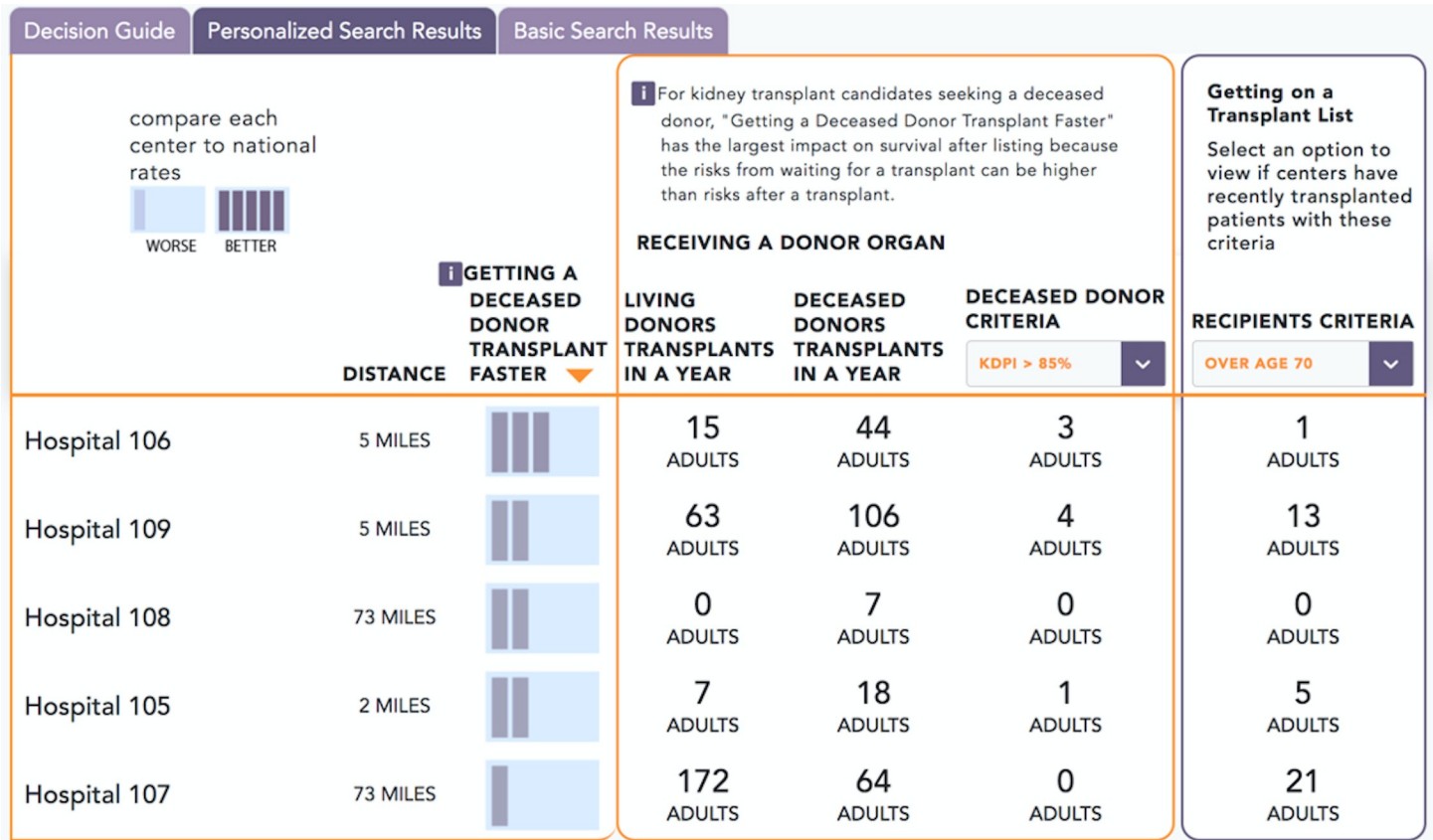

**Fig 5. Development website screenshot of a search results for a transplant center within a provided radius.** The page includes center outcomes and the characteristics of patients at a center and donors at a center.

feedback form to allow future users to submit comments, and future work can incorporate ongoing feedback.

The survey and usability testing demonstrated a feasible mixed methods approach to develop a patient-centered and data-driven website. The development concluded with a functioning website (www.transplantcentersearch.org) that is positively received by users and supports patient decisions when seeking an organ transplant with information tailored to patient characteristics. The mixed methods approach, combining survey and usability testing methods, provided flexibility during development to address multiple design questions including how graphic presentation of data influences decisions and how the content and user interface are interpreted by users. The approach is applicable for a broad range of target user groups who are faced with challenging decisions and a need for tailored information.

## Supporting information

**S1 Table. Full text of online survey questions and response options.**
(DOCX)

**S2 Table. Usability testing discussion guide questions.**
(DOCX)

**S3 Table. Usability testing health history questionnaire for kidney transplant patients.**
(DOCX)

**S4 Table. Selection of feedback from website users not enrolled in study.**
(DOCX)

**S1 Data. Raw survey data.**
(CSV)

## Author Contributions

**Conceptualization:** Marilyn J. Bruin, Ajay K. Israni, Cory R. Schaffhausen.

**Data curation:** Sauman Chu, Warren T. McKinney, Cory R. Schaffhausen.

**Formal analysis:** Sauman Chu, Cory R. Schaffhausen.

**Funding acquisition:** Ajay K. Israni.

**Methodology:** Sauman Chu, Cory R. Schaffhausen.

**Project administration:** Ajay K. Israni.

**Supervision:** Marilyn J. Bruin.

**Writing – original draft:** Sauman Chu, Cory R. Schaffhausen.

**Writing – review & editing:** Marilyn J. Bruin, Warren T. McKinney, Ajay K. Israni.

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
