## [Decision Letter · Decision Letter 0]

22 Mar 2021

PONE-D-21-00660

Design of a patient-centered decision support tool when selecting an organ transplant center

PLOS ONE

Dear Dr. Schaffhausen,

Thank you for submitting your manuscript to PLOS ONE. After careful consideration, we feel that it has merit but does not fully meet PLOS ONE’s publication criteria as it currently stands. Therefore, we invite you to submit a revised version of the manuscript that addresses the points raised during the review process.

The Section Editor and an Invited Reviewer evaluated the manuscript. There are several aspects of this manuscript which need to be addressed. The Section Editor and the Reviewer agree that the website is providing useful information for patients who are searching for a transplant center. Both also agree that such website is needed. However, such website needs to provide many additional information. The evaluation of the usefulness of the website should include reasons for the choice by the consumer (patient). Consequently, the following points need to be addressed: 

The Section Editor's comments: 

The informative website about available transplant centers is a very practical and useful tool for patients who are searching for a transplant center. Providing many information such as location, distance, number of transplants, specialized considerations (viral infection, diabetes and others) are alsovery helpful. Possibly, the website is under continuous development and the authors are considering added many more practical information such as the performance, additional services, and even accommodations. All these aspects are promising for the project. 

The manuscript needs to address the following points: 

The major problem of the report/manuscript about the website: (www.transplantcentersearch.org) is the fact that the authors are describing the functionality of this side without any information about the decision made by users. The entire benefit of this website is by providing name and location of transplant centers, the summary of the number of transplants, and the relative distance for the transplant centers. However, the website is following the rule of state-associated transplant centers, whereas the system is being transformed into the 250-mile zones. 

The authors had 684 participants influencing about 26 icon styles and 38 usability testing participants about some improvements to the website. These results are relatively limited and providing readers with no fundamental substance. The authors do not report what decisions were mage based on their website. What information influenced the decision making? What were the decisive points for the selection of specific transplant center? The manuscript does not even evaluate whether participants were transplant patients. Consequently, it is very difficult to evaluate what impact this website made except of intuitive suggestion that this type of website with information about the outcome provide some initial information. 

The authors need to combine their website usability with specific decision making by viewers. The description of website development and changes to icon style and other website aspect does not provide sufficient information about the impact on the choice of the transplant center. The data described are very preliminary and need to be further developed into more specific survey and concrete quality-related issues such as waiting time for a transplant, quality of service, availability of support personnel, outcomes of transplant results. All these factors need to be as materials available for transplant center evaluations. While the website informs which center has a shorter waiting time there is no details and therefore this may be insignificant in the context of the major issue to get a transplant. 

The manuscript does not provide practical and useful conclusions. 

The Reviewer's comments: 

The study by Schaffhausen et al describes the development of a website aimed at helping patients choose a transplant center. The study was performed in two stages: first, website icons were tested for various ways of displaying transplant centers as “good” or “bad”, second, various website structure versions were tested for functionality from a user perspective. Leaving out the question about the match between the research question and the scope of the journal, I would say this is a beautifully described study that deserves to be published. It seems backwards that the icon surveys were performed with regular internet users, while useability testing was performed with real waiting list patients. It may also be a politically charged question how an icon design can impact patient’s navigation and ultimate decision making, given that an icon is designed to represent transplant center ranking. None of this, however, should be a serious concern, at least not from my side

We look forward to receiving your revised manuscript.

Kind regards,

Stanislaw Stepkowski

Academic Editor

PLOS ONE

Journal Requirements:

2. Thank you for including your ethics statement:  "Usability testing: Each participant gave written, informed consent, and the study was approved by each institution’s Human Subjects Research Committee (HHS Study #16-4130 / UMNF Study #1606S89161).

Survey: Each participant consented to the survey electronically; written consent was waived because participants were anonymous.".   

3. In your Methods section, please provide a justification for the sample size used in your study, including any relevant power calculations (if applicable).

Additional Editor Comments:

The Section Editor and an Invited Reviewer evaluated the manuscript. There are several aspects of this manuscript which need to be addressed. The Section Editor and the Reviewer agree that the website is providing useful information for patients who are searching for a transplant center. Both also agree that such website is needed. However, such website needs to provide many additional information. The evaluation of the usefulness of the website should include reasons for the choice by the consumer (patient). Consequently, the following points need to be addressed:

The Section Editor's comments:

The informative website about available transplant centers is a very practical and useful tool for patients who are searching for a transplant center. Providing many information such as location, distance, number of transplants, specialized considerations (viral infection, diabetes and others) are alsovery helpful. Possibly, the website is under continuous development and the authors are considering added many more practical information such as the performance, additional services, and even accommodations. All these aspects are promising for the project. The manuscript needs to address the following points:

The major problem of the report/manuscript about the website: (www.transplantcentersearch.org) is the fact that the authors are describing the functionality of this side without any information about the decision made by users. The entire benefit of this website is by providing name and location of transplant centers, the summary of the number of transplants, and the relative distance for the transplant centers. However, the website is following the rule of state-associated transplant centers, whereas the system is being transformed into the 250-mile zones.

The authors had 684 participants influencing about 26 icon styles and 38 usability testing participants about some improvements to the website. These results are relatively limited and providing readers with no fundamental substance.

The authors do not report what decisions were mage based on their website. What information influenced the decision making? What were the decisive points for the selection of specific transplant center? The manuscript does not even evaluate whether participants were transplant patients. Consequently, it is very difficult to evaluate what impact this website made except of intuitive suggestion that this type of website with information about the outcome provide some initial information.

The authors need to combine their website usability with specific decision making by viewers. The description of website development and changes to icon style and other website aspect does not provide sufficient information about the impact on the choice of the transplant center. The data described are very preliminary and need to be further developed into more specific survey and concrete quality-related issues such as waiting time for a transplant, quality of service, availability of support personnel, outcomes of transplant results. All these factors need to be as materials available for transplant center evaluations. While the website informs which center has a shorter waiting time there is no details and therefore this may be insignificant in the context of the major issue to get a transplant.

The manuscript does not provide practical and useful conclusions.

The Reviewer's comments:

The study by Schaffhausen et al describes the development of a website aimed at helping patients choose a transplant center. The study was performed in two stages: first, website icons were tested for various ways of displaying transplant centers as “good” or “bad”, second, various website structure versions were tested for functionality from a user perspective. Leaving out the question about the match between the research question and the scope of the journal, I would say this is a beautifully described study that deserves to be published. It seems backwards that the icon surveys were performed with regular internet users, while useability testing was performed with real waiting list patients. It may also be a politically charged question how an icon design can impact patient’s navigation and ultimate decision making, given that an icon is designed to represent transplant center ranking. None of this, however, should be a serious concern, at least not from my side

Reviewers' comments:

Reviewer's Responses to Questions

**Comments to the Author**

1. Is the manuscript technically sound, and do the data support the conclusions?

Reviewer #1: Yes

Reviewer #2: No

2. Has the statistical analysis been performed appropriately and rigorously? 

Reviewer #1: N/A

Reviewer #2: No

3. Have the authors made all data underlying the findings in their manuscript fully available?

Reviewer #1: Yes

Reviewer #2: No

4. Is the manuscript presented in an intelligible fashion and written in standard English?

Reviewer #1: Yes

Reviewer #2: Yes

5. Review Comments to the Author

Reviewer #1: The study by Schaffhausen et al describes the development of a website aimed at helping patients choose a transplant center. The study was performed in two stages: first, website icons were tested for various ways of displaying transplant centers as “good” or “bad”, second, various website structure versions were tested for functionality from a user perspective. Leaving out the question about the match between the research question and the scope of the journal, I would say this is a beautifully described study that deserves to be published. It seems backwards that the icon surveys were performed with regular internet users, while useability testing was performed with real waiting list patients. It may also be a politically charged question how an icon design can impact patient’s navigation and ultimate decision making, given that an icon is designed to represent transplant center ranking. None of this, however, should be a serious concern, at least not from my side.

Reviewer #2: The informative website about available transplant centers is a very practical and useful tool for patients who are searching for a transplant center. Providing many information such as location, distance, number of transplants, specialized considerations (viral infection, diabetes and others) are alsovery helpful. Possibly, the website is under continuous development and the authors are considering added many more practical information such as the performance, additional services, and even accommodations. All these aspects are promising for the project. The manuscript needs to address the following points:

The major problem of the report/manuscript about the website: (www.transplantcentersearch.org) is the fact that the authors are describing the functionality of this side without any information about the decision made by users. The entire benefit of this website is by providing name and location of transplant centers, the summary of the number of transplants, and the relative distance for the transplant centers. However, the website is following the rule of state-associated transplant centers, whereas the system is being transformed into the 250-mile zones.

The authors had 684 participants influencing about 26 icon styles and 38 usability testing participants about some improvements to the website. These results are relatively limited and providing readers with no fundamental substance.

The authors do not report what decisions were mage based on their website. What information influenced the decision making? What were the decisive points for the selection of specific transplant center? The manuscript does not even evaluate whether participants were transplant patients. Consequently, it is very difficult to evaluate what impact this website made except of intuitive suggestion that this type of website with information about the outcome provide some initial information.

The authors need to combine their website usability with specific decision making by viewers. The description of website development and changes to icon style and other website aspect does not provide sufficient information about the impact on the choice of the transplant center. The data described are very preliminary and need to be further developed into more specific survey and concrete quality-related issues such as waiting time for a transplant, quality of service, availability of support personnel, outcomes of transplant results. All these factors need to be as materials available for transplant center evaluations. While the website informs which center has a shorter waiting time there is no details and therefore this may be insignificant in the context of the major issue to get a transplant.

The manuscript does not provide practical and useful conclusions.

6. PLOS authors have the option to publish the peer review history of their article (what does this mean?). If published, this will include your full peer review and any attached files.

Reviewer #1: **Yes: **Dulat Bekbolsynov

Reviewer #2: **Yes: **Stanislaw Stepkowski 

---

## [Author Response · Author response to Decision Letter 0]

12 Apr 2021

Response to reviewers

Manuscript PONE-D-21-00660

"Design of a patient-centered decision support tool when selecting an organ transplant center"

The Section Editor's comments: 

The informative website about available transplant centers is a very practical and useful tool for patients who are searching for a transplant center. Providing many information such as location, distance, number of transplants, specialized considerations (viral infection, diabetes and others) are also very helpful. Possibly, the website is under continuous development and the authors are considering added many more practical information such as the performance, additional services, and even accommodations. All these aspects are promising for the project.

RESPONSE: Thank you for your interest and suggested continuous improvements in this project. We agree that patients and families could benefit from information on additional services that are not currently included on the transplant center website. In our meetings with patients, we frequently heard similar suggestions. We do hope to make continuous improvements as we learn about how the site could better support patients. The data we use is provided by the Scientific Registry of Transplant Recipients (SRTR). Information such as additional services and accommodations is currently not available in the dataset. As the SRTR data evolves, we will continue to evolve our tool. For example, Hepatitis C positive donor transplants is a relatively new type of transplant and was not available a few years ago. We have now acknowledged this in the Discussion (Page 11, line 286)

The manuscript needs to address the following points: 

The major problem of the report/manuscript about the website: (www.transplantcentersearch.org) is the fact that the authors are describing the functionality of this side without any information about the decision made by users. 

RESPONSE: We agree that additional information on the decisions made by website users is important. The intention of the survey data was to understand how viewing different page styles could impact a decision, e.g. “what hospital would you choose for a transplant based on the information on this list?” Patients also discussed decision making during usability testing; however, this qualitative sample was not intended or powered to analyze decisions. The study has now begun a randomized trial of existing patients and members of the public screened for a likely future need for organ transplant (NCT03610555 on clinicaltrials.gov). The trial will measure patients’ decisions for a center and will compare the site described in the manuscript to an existing SRTR.org site. This trial is ongoing for 6-12 months and is beyond the scope of the current manuscript. We have clarified future work for patient decisions in the Discussion (page 14, line 363).

We are also collecting ongoing feedback from website visitors as to how this website is useful for their decision making. The summary of that finding is included in a new Supplement Table S4 (referenced on page 14, line 361).

The entire benefit of this website is by providing name and location of transplant centers, the summary of the number of transplants, and the relative distance for the transplant centers. However, the website is following the rule of state-associated transplant centers, whereas the system is being transformed into the 250-mile zones. 

RESPONSE: The recent change in liver allocation policy is relevant for the context of this work; however, several factors reinforce that the site remains a benefit. The search function includes two methods to enter a location, selecting a state or by entering a zip code and search radius. The latter approach could reflect the radius used in allocation policy and is available if a user prefers this method. A primary objective for the policy change was to influence donor allocation; therefore, the policy change is not targeting changes to waitlisting practices at a center. The website provides data to infer waitlisting practices, and this data could remain relevant, including a search result using state boundaries. Lastly, the data is updated twice per year, and as new policies impact transplant outcomes, the website data will reflect current practice. These points have been clarified in the Discussion (page 14, line 366).

The authors had 684 participants influencing about 26 icon styles and 38 usability testing participants about some improvements to the website. These results are relatively limited and providing readers with no fundamental substance. The authors do not report what decisions were made based on their website. What information influenced the decision making? What were the decisive points for the selection of specific transplant center? 

RESPONSE: The survey data included the following question, “What was the most important factor in the decision?”, see Table 2 and Supplement Table S1.

We are also collecting ongoing feedback from website visitors, including candidates and recipients, as to how this website is useful for their decision making. The summary of that finding is in a new Supplemental Table S4 (referenced on page 14, line 361).

The manuscript does not even evaluate whether participants were transplant patients. Consequently, it is very difficult to evaluate what impact this website made except of intuitive suggestion that this type of website with information about the outcome provide some initial information. 

RESPONSE: The manuscript includes data from both patients and the general public, and both cohorts were evaluated whether or not they were patients. Survey data for the general public included the following question, “Have you ever needed or received an organ or other transplant?”, see Table 1 and Supplement Table S1. This has been clarified in the Methods (page 5, line 123). The study intentionally included data from respondents who did not previously receive a transplant because individuals who have recently learned of a potential future need for a transplant have likely not received significant education about listing practices at centers and have knowledge similar to the general public. In our previous work, even transplant candidates were often unaware of differences in listing practices (Liver Transplantation. 2020;26(3):337-48. Transplantation Direct. 2020;6(8):e585-e). In addition, family members who may have less direct experience with transplant providers may also assist in decision making. This has been clarified in the Discussion (page 14, line 378). The user-centered and mixed methods approach to develop the website is described in the manuscript, and the randomized trial for effectiveness is ongoing, future work and beyond the scope of website development.

The authors need to combine their website usability with specific decision making by viewers. The description of website development and changes to icon style and other website aspect does not provide sufficient information about the impact on the choice of the transplant center. The data described are very preliminary and need to be further developed into more specific survey and concrete quality-related issues such as waiting time for a transplant, quality of service, availability of support personnel, outcomes of transplant results. All these factors need to be as materials available for transplant center evaluations. While the website informs which center has a shorter waiting time there is no details and therefore this may be insignificant in the context of the major issue to get a transplant. 

RESPONSE: We agree that additional testing of the existing website is warranted, and this trial is ongoing. The Discussion (page 14, line 363) more clearly describes this future work. As described above, the scope of the current project was intentionally limited to include data that currently exists in a national database. As the data available evolves, the website will evolve. Therefore, the website states the date of the last data available. We have clarified this scope in the Discussion (page 11, line 286). 

The manuscript does not provide practical and useful conclusions.

RESPONSE: We have revised the conclusion (page 15, line 385)

The Reviewer's comments: 

The study by Schaffhausen et al describes the development of a website aimed at helping patients choose a transplant center. The study was performed in two stages: first, website icons were tested for various ways of displaying transplant centers as “good” or “bad”, second, various website structure versions were tested for functionality from a user perspective. Leaving out the question about the match between the research question and the scope of the journal, I would say this is a beautifully described study that deserves to be published. 

RESPONSE: We appreciate your feedback on the strengths of the study.

It seems backwards that the icon surveys were performed with regular internet users, while useability testing was performed with real waiting list patients. 

RESPONSE: A rationale for this design was to measure decision making using a sample powered for this analysis. For a series of studies that takes place during website development, recruiting the same number of patients for the survey was likely not feasible. We have previously recruited patients for surveys and the results were smaller numbers (Clinical Transplantation. 2018;32:e13426.). However, a smaller sample size of transplant patients was sufficient to provide qualitative feedback. This rationale is included in the Discussion (page 14, line 378). 

It may also be a politically charged question how an icon design can impact patient’s navigation and ultimate decision making, given that an icon is designed to represent transplant center ranking. None of this, however, should be a serious concern, at least not from my side

RESPONSE: We appreciate the understanding of potential politically changed points of view. The current study focused on patient and public decisions and perspectives as this was the priority for a public-facing website. The methods were consistent with guidelines on developing public healthcare reports (J. Hibbard and S. Sofaer. How to effectively present health care performance data to consumers. Institution: Agency for Healthcare Research and Quality. 2010. Rockville, MD)

---

## [Editor Report · Decision Letter 1]

20 Apr 2021

Design of a patient-centered decision support tool when selecting an organ transplant center

PONE-D-21-00660R1

Dear Dr. Schaffhausen,

We’re pleased to inform you that your manuscript has been judged scientifically suitable for publication and will be formally accepted for publication once it meets all outstanding technical requirements.

Kind regards,

Stanislaw Stepkowski

Academic Editor

PLOS ONE

Additional Editor Comments (optional):

None
---

## [Editor Report · Acceptance letter]

5 May 2021

PONE-D-21-00660R1 

Design of a patient-centered decision support tool when selecting an organ transplant center 

Dear Dr. Schaffhausen:

I'm pleased to inform you that your manuscript has been deemed suitable for publication in PLOS ONE. Congratulations! Your manuscript is now with our production department. 

Kind regards, 

on behalf of

Dr. Stanislaw Stepkowski 

Academic Editor

PLOS ONE